# Serum wisteria floribunda agglutinin-positive human Mac-2 binding protein is unsuitable as a diagnostic marker of occult hepatocellular carcinoma in end-stage liver cirrhosis

Kantoku Nagakawa, Masaaki Hidaka, Takanobu Hara, Hajime Matsushima, Hajime Imamura, Takayuki Tanaka, Tomohiko Adachi, Akihiko Soyama, Kengo Kanetaka, Susumu Eguchi[ID]*

Department of Surgery, Nagasaki University Graduate School of Biomedical Sciences, Nagasaki, Nagasaki, Japan

* sueguchi@nagasaki-u.ac.jp

## Abstract

### Background and purpose

Serum glycosylated *Wisteria floribunda* agglutinin-positive Mac-2 binding protein (WFA+-M2BP) is a marker of liver fibrosis and hepatocellular carcinoma (HCC). In this study, we aimed to evaluate the diagnostic ability of WFA+-M2BP for occult HCC, which current diagnostic imaging tests fail to detect.

### Methods

Patients who underwent hepatectomy for liver transplantation (LT) and whose whole liver could be sliced and subjected to histological examination between 2010 and 2018 were eligible for this study (n = 89). WFA+-M2BP levels were measured in samples collected before the LT. Comparison of the postoperative histological test results with the preoperative imaging data grouped the patients into histologically no group (N), histologically detected group (D), histologically increased group (I), and histologically decreased or same group (DS), and the results were compared with the WFA+-M2BP values. In addition, comparisons were made between each data with and without HCC, including occult HCC, and total tumor diameter.

### Results

Irrespective of underlying hepatic disease conditions, there were 6 patients in the N group, 10 in the D group, 41 in the I group, and 32 in the DS group. The median of the serum WFA+-M2BP level for each group was as follows: N group, 8.05 (1.25–11.9); D group, 11.025 (1.01–18.21); I group, 9.67 (0.29–17.83); and DS group, 9.56 (0.28–19.44) confidence of interval. We found no significant differences between the pairings. Comparison of underlying hepatic diseases revealed that liver cirrhosis due to hepatitis B and C and non-B and -C liver cirrhosis had no significant differences. AFP levels, on the other hand, had

**Data Availability Statement:** All relevant data are within the paper and its Supporting Information files.

**Funding:** The authors received no specific funding for this work.

**Competing interests:** The authors have declared that no competing interests exist.

significant relationships in comparison between the presence or absence of histological HCC, in correlation between total tumor diameter, and in the ROC analysis for the diagnosis of HCC including occult HCC.

## Conclusion

Serum WFA+-M2BP cannot help diagnose occult HCC that is already undetected using imaging tests in decompensated liver cirrhosis patients requiring LT.

## Introduction

Mac-2 binding protein (M2BP) is a marker of liver fibrosis and is used to diagnose liver cirrhosis. M2BP is a glycoprotein secreted by various cells, including hepatocytes, and regulates many mechanisms like cell adhesion [1, 2]. Levels of isomers with glycan structures that bind to specific lectins (M2 and BP) increase with progressing liver fibrosis. It has been reported that the amount of M2BP attached to *Wisteria floribunda* agglutinin (WFA) significantly positively correlates with the degree of liver fibrosis [1] and acts a predictor of liver cirrhosis and hepatocellular carcinogenesis in chronic hepatitis C cases [2, 3]. Furthermore, in chronic hepatitis B patients with hepatocellular carcinoma (HCC) diagnosed using imaging tests, a significant increase in M2BP is observed, suggesting the possibility of early diagnosis using M2BP level determination [4]. We performed living donor liver transplantation (LT) for patients with decompensated cirrhosis and histological diagnosis of the whole liver removed from the recipient during LT. We have previously often detected the presence of small and occult HCCs in removed livers, which cannot be detected using preoperative ultrasonography or computed tomography (CT), indicating a limitation to cancer diagnosis using existing preoperative tests. Although HCC is diagnosed using blood tumor markers, alpha-fetoprotein (AFP), protein induced by vitamin K absence-II (PIVKA-II), and imaging tests such as ultrasonography and CT, current tests still fail to detect minute HCCs. M2BP level being positively correlated with the degree of HCC could be an important marker for detecting HCC. The aim of this study was to clarify whether the abovementioned glycosylated isomers of M2BP was able to be used as markers for early diagnosis of occult HCC that remains undiagnosed using pre-existing blood tests or imaging tests.

## Materials and methods

### Patients

We evaluated 89 patients who underwent LT at Nagasaki University Hospital between 2010 and 2018. The recipients were > 18 years old with various liver diseases, such as liver cirrhosis associated with hepatitis B or C virus or alcohol, and HCC. We evaluated the patients' blood test results for α-fetoprotein (AFP) and protein induced by vitamin K antagonists-II (PIVKA-II), scoring of liver function, Child-Pugh score and Model for end-stage liver disease (MELD) score, and histological diagnosis of the whole liver removed during LT. Imaging study-based preoperative diagnostic criteria of HCC are based on the presence of focal hepatic lesions with hyperattenuation in the arterial phase and hypoattenuation in the portal phase on dynamic CT and Gd-EOB-DTPA-enhanced magnetic resonance imaging (EOB-MRI). These imaging tests were performed two to three weeks before LT. All of the tumors detected by CT were detected by MRI. There were the tumors weren't able to detected by CT but were detected

by MRI. So, we determined tumors weren't diagnosed by MRI as occult HCC. We compared the number of cancers diagnosed by histological examination with those obtained by imaging examination. The exclusion criteria were co-infection with hepatitis C and B viruses.

### Whole liver histological examination

As reported previously, after liver explantation, the cirrhotic livers were fixed in formalin for 48 h and then sliced into 5–7 mm cubes [5, 6]. After careful mapping, they were embedded in paraffin and stained with hematoxylin and eosin. Experienced pathologists examined all the slides.

### Measurement of WFA$^+$-M2BP

WFA$^+$-M2BP levels were measured as previously described [2]. WFA$^+$-M2BP level was quantified based on a lectin-antibody sandwich immunoassay using a fully automatic immunoanalyzer (HISCL-2000i, Sysmex, Hyogo, Japan). The measured values of WFA+-M2BP conjugated to WFA were calculated using the following equation: cut-off index (COI) = ([WFA$^+$-M2BP]$_{sample}$—[WFA$^+$-M2BP]$_{negative\ controls}$) / ([WFA$^+$-M2BP]$_{positive\ controls}$—[WFA$^+$-M2BP]$_{negative\ controls}$).

### Statistical analysis

Data were expressed as median values (range) as appropriate. Comparisons between the groups were done using the Wilcoxon signed-rank test. Receiver operating characteristic curves were constructed, and the areas under the curves were compared to test the ability to diagnose occult HCC. Optimal cut-off values for the different scales were determined the Youden's index, which is the value that maximizes the sum of the sensitivity and specificity. Statistical significance was set at p <0.05. Statistical analyses were performed using JMP Pro 16.0.0 software (SAS Institute Inc., Cary, NC, USA).

## Results

### Occult HCC diagnosed by EOB-MRI

Two hundred eleven tumors were able to be counted as that were well-circumscribed and measurable in all patients. There were 152 well-differentiated HCCs and 59 moderately-differentiated HCCs. Of 152 well-differentiated hepatocellular carcinomas, 42 tumors were able to be detected by EOB-MRI. Of 59 moderate-differentiated hepatocellular carcinomas, 39 tumors were able to be detected by EOB-MRI. The data for each tumor diameter are shown in the Table 1.

**Table 1. The histological differentiation and the diagnostic ability of EOB-MRI.**

| Histological differentiation | The number of tumors | The diameter Median (mm) | Significant difference | Logistic regression The size of detectable limit |
|---|---|---|---|---|
| **well-differentiated HCC 152** | Detected tumor 42 | 15 (4–45) | p<0.0001 | Cut-off value 9 mm AUC 0.922 |
| | Undetected tumor 110 | 4 (1–15) | | |
| **moderately-differentiated HCC 59** | Detected tumor 39 | 18 (8–55) | p<0.0001 | Cut-off value 12 mm AUC 0.964 |
| | Undetected tumor 20 | 8 (3–12) | | |

EOB-MRI; Gd-EOB-DTPA-enhanced magnetic resonance imaging.

### Serum WFA+-M2BP in all patients correlated with Child-Pugh and Meld scores but not with tumor markers

The median serum level of the WFA[+]-M2BP index in all patients was 9.53 (0.28–19.44) COI. The WFA[+]-M2BP index was correlated with the Child-Pugh score (r = 0.50, p<0.01) and MELD score (r = 0.31, p<0.01). It had no co-relationship with the tumor markers AFP and PIVKA-II.

Postoperative histological examination diagnosed 83 patients with HCC, 10 in whom HCC was not detected during preoperative imaging. In 41 patients, the number of tumors was higher than that diagnosed during a preoperative imaging test, and 32 had the same or fewer tumors than that diagnosed by the preoperative imaging test. Six patients were undiagnosed with HCC during preoperative imaging or postoperative histological tests. We divided the patients into four groups: 1) group in which HCC was not detected during both preoperative imaging test and postoperative histological test; none group (N); 2) group in which HCC was detected during postoperative histological test but not preoperative imaging test; detected group (D); 3) group in which a preoperative imaging test detected HCC, and the number increased in the postoperative histological test; increased group (I); and 4) group in which a

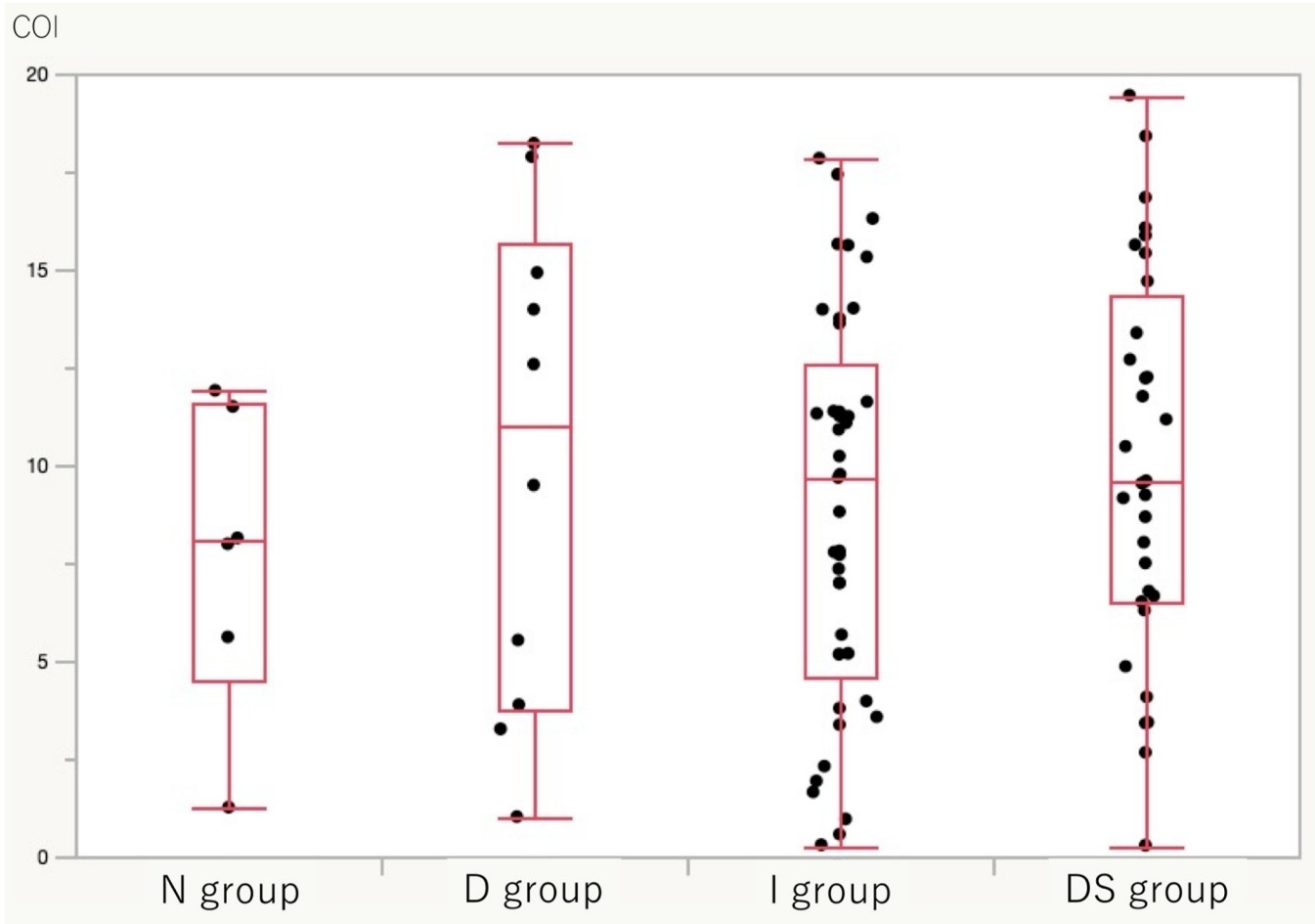

**Fig 1. Comparison of the M2BP index of each group regardless of underlying hepatic diseases.** No significant difference was observed between the pairs. The difference between the N and D groups can be used to assess the ability to detect occult HCC. However, there was no significant difference (N group vs D group: 8.055 vs 11.025, p = 0.705).

preoperative imaging test detected HCC, and the number was the same or decreased in the postoperative histological test; decreased or same group (DS). This grouping allowed us to evaluate the effectiveness of the M2BP index for detecting occult HCC. The M2BP index was considered effective if there was a significant difference between groups D and N.

## Serum WFA[+]-M2BP level does not predict presence of HCC

Regardless of underlying hepatic diseases, the N group had 6 patients, the D group had 10 patients, the I group had 41 patients, and the DS group had 32 patients. The median of the serum WFA[+]-M2BP level of each group was as follows: N group 8.05 (1.25–11.9), D group 11.03 (1.01–18.21), I group 9.67 (0.29–17.83), and DS group 9.56 (0.28–19.44) COI. There was no significant difference between the pairings. (Fig 1)

## AFP may serve as a biomarker for HCC

There were no significant differences in Child-Pugh and MELD scores. Regarding tumor markers, while the value of PIVKA-II showed no significant difference, two pairs had significant differences in AFP levels. The values of AFP of N group vs I group were 4.45 (0.9–22.6) vs 22.55 (0.8–1569) ng/mL (p = 0.03) and N group vs DS group were 4.45 (0.9–22.6) vs 31.8 (1.6–993.7) ng/mL (p = 0.01). (Table 2 and Fig 2) Excluding this grouping and comparing the existence of histological HCC and total tumor diameter with M2BP index or tumor markers, there was a significant difference only in AFP values with and without HCC. In comparison of total tumor diameter with each data, there was no significant difference in M2BP index and positive correlation between total tumor diameter and AFP, or PIVKA-II. In addition, the ROC curve analysis showed that the area under the curve of AFP was 0.7757, and that cut-off value was 23.20. (Table 3)

**Table 2. Characteristics of each group.**

|  | N group | D group | I group | DS group |
|---|---|---|---|---|
| **Case number** | 6 | 10 | 41 | 32 |
| **Age** | 57 (52–62) | 54 (33–69) | 60 (48–71) | 60.5 (52–72) |
| **Sex (M : F)** | 3 : 3 | 5 : 5 | 31 : 10 | 19 : 13 |
| **HBV** | 1 | 2 | 7 | 5 |
| **HCV** | 3 | 6 | 27 | 14 |
| **Co-infection HBV/HCV** | 0 | 1 | 2 | 0 |
| **non B, non C** | 2 | 1 | 5 | 13 |
| **Child-Pugh score** | 9 (8–12) | 9 (6–14) | 10 (5–13) | 10 (7–13) |
| **MELD score** | 14.5 (11–30) | 13 (7–29) | 13 (4–31) | 14 (7–26) |
| **fibrosis factor #** | 4 | 4 | 4 | 4 |
| **Total tumor diameter (mm)** | 0 | 10.5 (7–22) | 40.5 (9–106) | 25 (1–58) |
| **UICC Stage** | - | 1A = 5<br>2 = 5 | 2 = 41 | 1A = 9<br>1B = 6<br>2 = 16<br>3A = 1 |
| **AFP (ng/mL)** | 4.45 (0.9–22.6) | 9.6 (2.4–50.8) | 22.55 (0.8–1569) | 31.8 (1.6–993.7) |
| **PIVKA-II (mAU/ml)** | 160 (13–569) | 21.5 (11–175) | 63.5(10–13910) | 95 (10–3091) |
| **M2BP index** | 8.05 (1.25–11.9) | 11.03 (1.01–18.21) | 9.67 (0.29–17.83) | 9.56 (0.28–19.44) |

HBV, hepatitis B virus; HCV, hepatitis C virus; MELD score, a model for end-stage liver disease score; UICC, Union for International Cancer Control; AFP, α-fetoprotein; PIVKA-II, protein induced by vitamin K absence or antagonists-II; M2BP, Mac-2 binding protein, # Histological fibrosis factor from New Inuyama Classification.

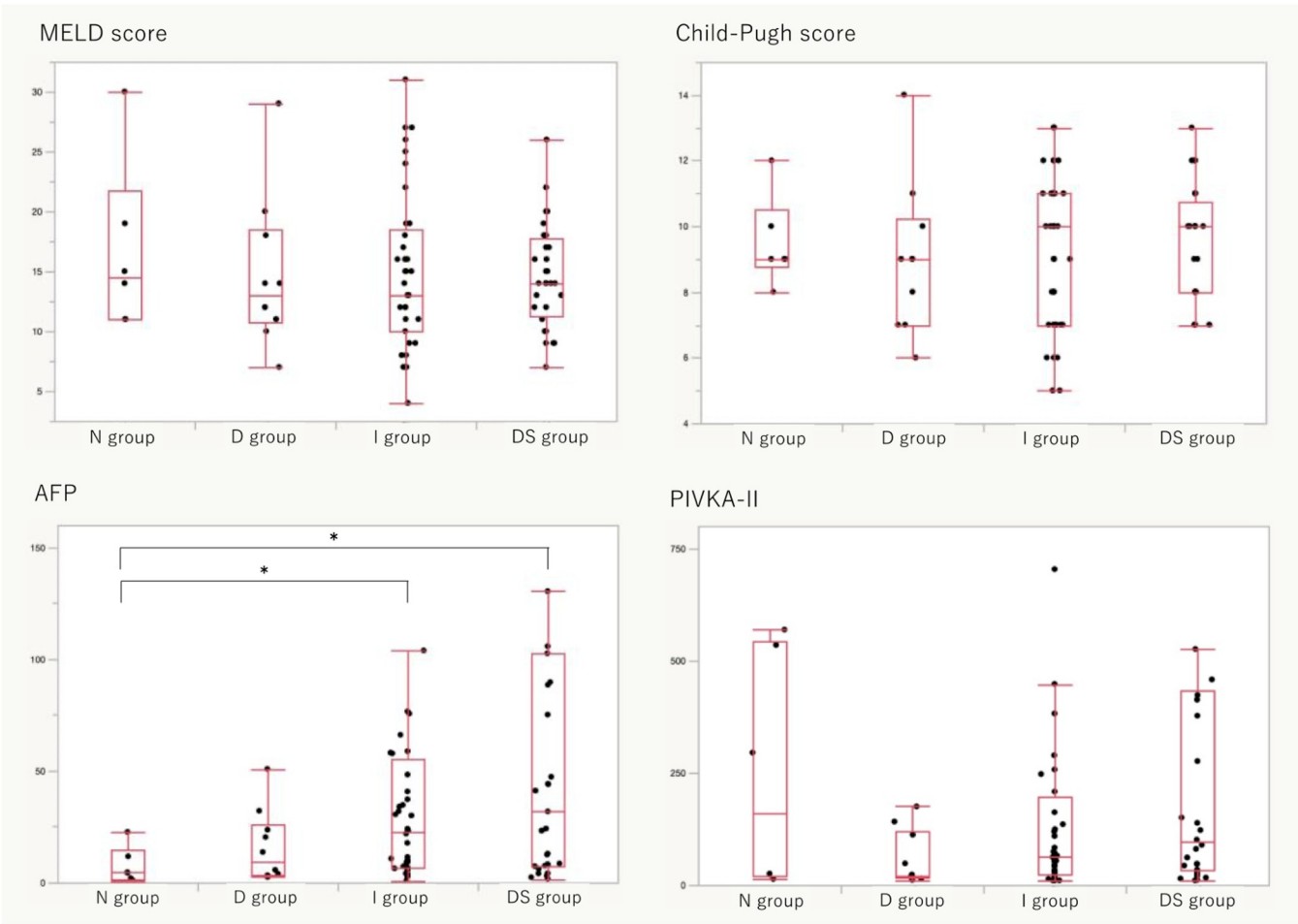

**Fig 2. Comparison of the MELD score, Child-Pugh score, and tumor markers of each group regardless of underlying hepatic diseases.** * Significant difference Only AFP pairs (N group vs I group and N group vs DS group) showed significant differences. However, there was no significant difference between the N and I groups, even in AFP levels. None of the values could be used for the detection of occult HCC.

### M2BP index did not predict presence of HCC in each underlying hepatic disease

We compared the data in the same way for each of the underlying hepatic diseases. In liver cirrhosis due to hepatitis B and C virus, and non-B and non-C hepatitis, each value of the M2BP index showed no significant difference. In addition, there were no significant differences in MELD scores. No significant difference was found in the presence of tumor markers with the exception of AFP levels in hepatitis C. Significant differences in AFP levels were seen between groups N vs DS (4.6 [1.9–11.7] vs 47.3 [4.1–956.7] ng/mL [p = 0.03]) and groups D vs DS (16.9 [3.8–32.1] vs 47.3 [4.1–956.7] ng/mL [p = 0.04]). (Fig 3)

## Discussion

It is evident that a very early diagnosis of HCC can lead to a good prognosis through early treatment with various modalities. In addition, the size and number of HCC are related to the indication of liver transplantation by the Milan criteria [7]. It is therefore important to diagnose occult HCC for a success outcome of liver transplantation. Diagnosis of occult HCC relies

**Table 3. The diagnostic ability of HCC.**

**Comparison between the presence or absence of histological HCC and each data**

|  | the presence of HCC | the absence of HCC | p value |
|---|---|---|---|
| M2BP index | 9.59 (0.28–19.44) | 8.05 (1.25–11.9) | 0.481 |
| AFP (ng/mL) | 23.2 (0.8–1569) | 4.4 (0.9–22.6) | 0.024 |
| PIVKA-II (mAU/ml) | 63.5 (10–13910) | 160 (13–569) | 0.741 |

**Correlation between total tumor diameter and each data**

|  | correlation coefficient | p value |
|---|---|---|
| M2BP index | -0.059 | 0.609 |
| AFP | 0.253 | 0.028 |
| PIVKA-II | 0.352 | 0.002 |

**The ROC curve analysis for the diagnosis of HCC including occult HCC**

|  | Cut-off value | AUC | Sensitivity | Specificity | p-value |
|---|---|---|---|---|---|
| M2BP index | 12.21 | 0.5863 | 0.3253 | 1.0000 | 0.400 |
| AFP (ng/mL) | 23.20 | 0.7757 | 0.5062 | 1.0000 | 0.013 |
| PIVKA-II (mAU/ml) | 27.00 | 0.4593 | 0.7250 | 0.5000 | 0.616 |

M2BP: Mac-2 binding protein, AFP: α-fetoprotein, PIVKA-II, protein induced by vitamin K absence or antagonists-II, AUC: Area under the curve.

on imaging tests. It is reported that EOB-MRI is superior to MD-CT in detecting small HCC [8]. In liver cirrhosis patients, however, the detectability of HCC is reduced in EOB-MRI due to the reduced contrast effect [9]. To this end, this study investigated whether identifying the WFA[+]-M2BP levels can overcome the detection limit of imaging tests for HCC diagnosis. One of the strengths of our study was that we could histologically examine the whole liver removed at the time of LT, which enabled us to detect occult HCC that were not detected in imaging tests.

WFA[+]-M2BP is secreted by hepatic stellate cells (HSCs) and enhances the synthesis of Mac-2 by Kupffer cells, which renders HSCs fibrogenic [10]. Hence, WFA[+]-M2BP serum levels can be a useful biomarker to identify the degree of liver fibrosis. In addition, several studies have reported that WFA[+]-M2BP levels help diagnose and predict the recurrence of HCC in HBV- and HCV-infected patients or patients with non-alcoholic fatty liver disease [3, 4, 11–13]. Patients with chronic viral hepatitis, regardless of the degree or severity of fibrosis, have elevated serum levels of WFA[+]-M2BP, indicating an increased occurrence of HCC [2, 14]. A study on liver stiffness also indicated that M2BP might be associated with the development of HCC, independent of fibrogenesis. In the study, the authors concluded serum WFA+-M2BP represented a sensitive marker of early-stage HCC detection in patients with chronic HBV infection [4]. This conclusion motivated us to investigate WFA[+]-M2BP as a detecting and diagnostic marker of occult HCC.

Our study showed that the WFA[+]-M2BP value could not help detect occult HCC that was undetected by imaging examination. It is the same result in the comparison with the addition of the cases to 30 in N group that were diagnosed as no HCC with histological test not using this study's protocol. We hypothesized that such an occurrence could be attributed to the fact that the liver in the target group was an end-stage cirrhotic liver, and the WFA[+]-M2BP value was detected as the degree of fibrosis. The incidence of HCC increases in patients with HBV hepatitis when the WFA[+]-M2BP value is above 0.69–1.8 [14–17], in patients with HCV hepatitis when the WFA[+]-M2BP value is above 1.75–4.2 [13, 18–20], and in NAFLD/NASH patients when the WFA[+]-M2BP value is above 1.255 [12]. In this study, the median value was 9.53 (0.28–19.44) for all patients and 8.05 (1.25–11.9) for non-carcinoma patients, which was

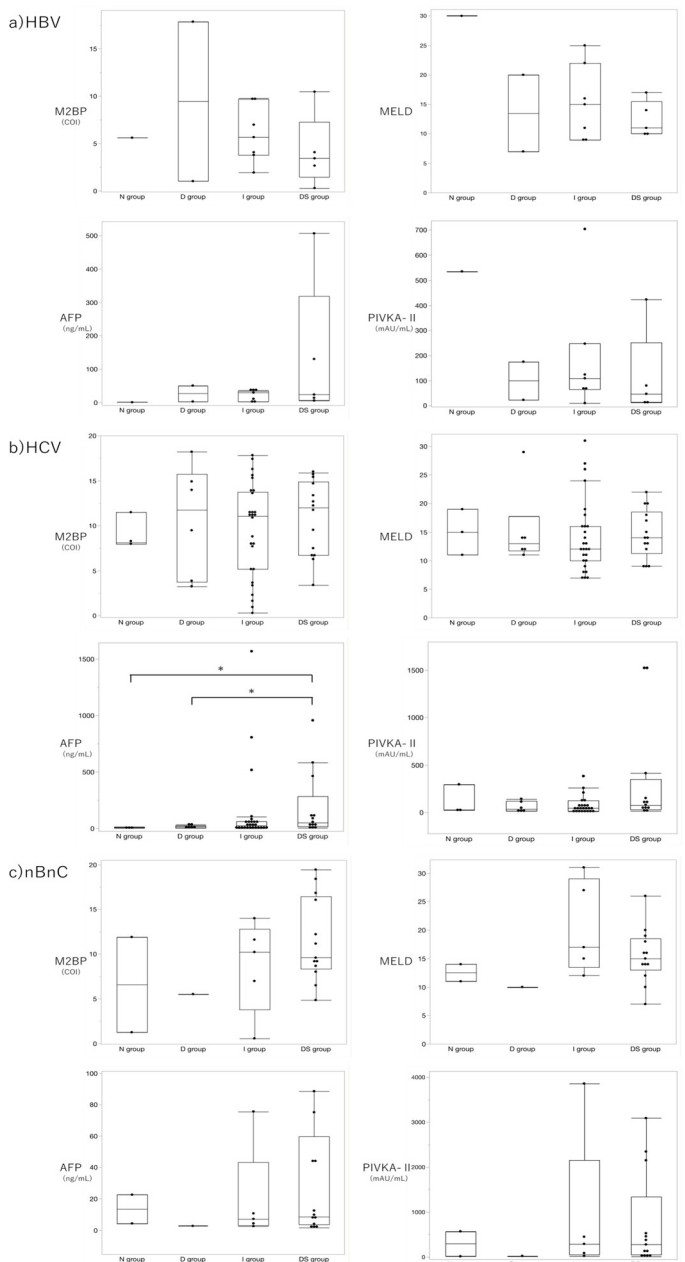

**Fig 3. Comparison of the M2BP index, hepatic reserve, and tumor markers of each group for each underlying hepatic disease.** * Significant difference No significant difference in the M2BP index of all underlying hepatic diseases was observed between the N and D groups. In patients with hepatitis C, there was a significant difference in the AFP values between group N vs DS and D vs DS. However, there were no significant difference in the AFP value between the N and D groups.

significantly higher than that in the previous studies which led us to recognize the strong fibrotic effect of liver. This may be attributed to the fact that the patients who require LT have decompensated cirrhosis. It was reported in 2002 that some cancer cells expressed M2BP, shedding light on its origin [21]. However, recently, hepatocytes and HSCs have been thought to secrete M2BP [10, 22]. In particular, it has been reported that M2BP-messenger RNA (mRNA) transcription in the fibrotic liver is activated in HSCs. M2BP itself is expressed in

Kupffer cells [22]. M2BP-mRNA expression was also correlated with serum WFA[+]-M2BP levels. The specificity of WFA[+]-M2BP for HCC is unclear because WFA[+]-M2BP is a detector index of glycosylation site binding to WFA lectin. If we suppose that M2BP, independent of fibrosis and associated with the development of HCC as suggested by previous studies, can be detected, then it may be possible to detect small HCCs using this research method.

Interestingly, the level of AFP tended to differ between groups; in some groups, the level of AFP was significantly higher than that in the N group. Considering only the presence of tumor or total tumor diameter, rather than this grouping, only AFP had significance in all statistical analysis. This suggests that AFP may be better diagnostic biomarker for occult HCC than WFA+-M2BP. However, we found no significant difference in AFP levels between the N and D groups, indicating a limit of detection. New biomarkers, including DNA, RNA, and protein biomarkers, such as AFP fraction L3, and conventional laboratory metrics for HCC diagnosis were reviewed by Wang et al. [23]; however, M2BP was not included in the review. Further study is needed to develop early detection methods for occult HCC.

Several limitations to the present study must be noted. The small number of cases in retrospective observational analyses in a single center was one of the limitations. Second, the number of patients with no HCC was small. However, we believe that our data are essential for whole liver histological examination with sectioned small specimens.

In conclusion, serum WFA[+]-M2BP is not suitable to diagnose occult HCC that had not been detected by imaging tests in decompensated liver cirrhosis patients who needed LT.

## Supporting information

**S1 File. Reference data.**
(DOCX)

## Author Contributions

**Conceptualization:** Kantoku Nagakawa, Masaaki Hidaka, Susumu Eguchi.

**Data curation:** Kantoku Nagakawa, Masaaki Hidaka, Takanobu Hara, Hajime Matsushima, Hajime Imamura, Takayuki Tanaka, Tomohiko Adachi.

**Funding acquisition:** Susumu Eguchi.

**Investigation:** Kantoku Nagakawa.

**Methodology:** Kantoku Nagakawa, Tomohiko Adachi, Akihiko Soyama.

**Project administration:** Kantoku Nagakawa, Akihiko Soyama.

**Supervision:** Tomohiko Adachi, Kengo Kanetaka, Susumu Eguchi.

**Writing – original draft:** Kantoku Nagakawa.

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
