## [Decision Letter · Decision Letter 0]

18 Jun 2023

PONE-D-23-05749Serum Wisteria floribunda agglutinin-positive human Mac-2 binding protein is unsuitable as a diagnostic marker of occulthepatocellular carcinoma in end-stage liver cirrhosis.PLOS ONE

Dear Dr. Nagakawa,

Thank you for submitting your manuscript to PLOS ONE. After careful consideration, we feel that it has merit but does not fully meet PLOS ONE’s publication criteria as it currently stands. Therefore, we invite you to submit a revised version of the manuscript that addresses the points raised during the review process.

We look forward to receiving your revised manuscript.

Kind regards,

Jincheng Wang

Academic Editor

PLOS ONE

2. Please include a copy of Table 1which you refer to in your text on page 9.

Reviewers' comments:

Reviewer's Responses to Questions

**Comments to the Author**

1. Is the manuscript technically sound, and do the data support the conclusions?

Reviewer #1: Yes

Reviewer #2: No

Reviewer #3: No

Reviewer #4: Partly

Reviewer #5: Partly

2. Has the statistical analysis been performed appropriately and rigorously? 

Reviewer #1: Yes

Reviewer #2: No

Reviewer #3: No

Reviewer #4: I Don't Know

Reviewer #5: No

3. Have the authors made all data underlying the findings in their manuscript fully available?

Reviewer #1: Yes

Reviewer #2: No

Reviewer #3: Yes

Reviewer #4: Yes

Reviewer #5: No

4. Is the manuscript presented in an intelligible fashion and written in standard English?

Reviewer #1: Yes

Reviewer #2: Yes

Reviewer #3: Yes

Reviewer #4: Yes

Reviewer #5: Yes

5. Review Comments to the Author

Reviewer #1: This paper by Nakagawa et al. clarified the clinical utility of serum Wisteria floribunda agglutinin-positive Mac-2 binding protein (M2BP) in end-stage liver disease. Using the explanted end-stage liver of liver transplant recipients, the authors showed that M2BP is not elevated due to occult hepatocellular carcinoma (HCC). The clinical usefulness of M2BP is evident in diagnosing liver fibrosis grade and assessing the risk of future development of HCC. However, the M2BP utility for evaluating the current hepatocellular carcinoma existence is controversial. This paper clarifies this point and is clinically meaningful. There are several comments from the reviewer.

Major

1. The main result in this paper is a comparison between N (who did not have HCC in the explanted liver) and D or I (who had more HCC than preoperatively diagnosed). However, as the author listed in the limitation section, the number of N groups (n=6) control is small as a control group. This may be a critical problem, so we must treat the results carefully.

2. To show that M2BP is not useful for assessing HCC existence, comparing the difference of M2BP between the pathological stage of HCC diagnosed using the whole slice explanted liver may be helpful.

Reviewer #2: This paper was written about WFA-M2BP and HCC. The usefullnes to detect HCC that was not shown preoepratively was studied. However, authors did not understand that the value of WFA-M2BP differs by the etiology of underlying liver disease. Therefore, This paper is wrong from the study design. Also, WFA-M2BP is not tumor marker but fibrosis marker and predictive marker.

Reviewer #3: Nagakawa K et al. demonstrated that serum WFA+-M2BP level was not associated with the presence of occult HCC in the explanted liver in liver transplantation. Although it is clinically important to detect HCC before liver transplantation, the authors should consider the following problems.

Major comment

1. The abstract seems confusing because the definition of each group is unclear in the abstract.

2. Many reports has been demonstrated the usefulness of WFA+-M2BP as a predictive marker of HCC development, but not as a diagnostic marker of HCC. The authors should refer to more references which demonstrated usefulness of WFA+-M2BP as a diagnostic marker of HCC.

3. The diagnostic ability of WFA+-M2BP and the other markers should be analyzed and compared by ROC analysis.

Reviewer #4: Nakagawa et al. reported that M2BPGi is unsuitable as a diagnostic marker of occult HCC end-stage liver cirrhosis. This paper is interesting in that it provides a detailed analysis of HCC in the liver of a liver transplant recipient. However, it appears that further analysis is needed to determine if M2BPGi can be used to detect occult HCC for analysis.

1) Since it is widely known that M2BPGi is a marker that also reflects fibrosis and inflammation, multiple measurements of M2BPGi seem more necessary for observation of HCC than a single measurement of M2BPGi.

2) On the empirical side, AFP, AFP-L3, and DCP may be better at detecting occult HCC. However, we believe that this also makes sense to measure over time.

3) HCC is a very heterogeneous group, so if you look at the individual cases, is there anyone who could have predicted occult HCC at all?

4) What is the best way to detect occult HCC based on this study by the author or in the literature?

Reviewer #5: K. Nagakawa et al. have reported about utility of serum WFA+-M2BP in occult hepatocellular carcinoma. It is interesting to investigate many patients with liver transplant, however there are serious problems with the study design and logical progress. There are several concerns in this study to be published as a research article.

Major

#1

While the purpose or motivation for the study to determine utility of WFA+-M2BP in occult HCC was understandable, it was ultimately not useful.

From the context, it was not a study design to show the negative data of WFA+-M2BP.

It would be more logical to reconsider from the viewpoint of finding occult HCC with other significant markers such as AFP, instead of forcing WFA+-M2BP to be the main objective.

#2

As the authors indicate, it was presumed that WFA+-M2BP was not significant when compared in disease groups at high levels WFA+-M2BP. Authors may investigate to adjust and to stratify by background liver fibrosis status (e.g; chronic hepatitis, / cirrhosis, fibrosis index).

＃3

The patients are divided in groups as N, D, I, and DS, however there was no Table which summarize each characteristic. Authors should summarize about characteristics of each group (e.g., age, gender, liver function: Bil, Alb, Fib-4 index, number or diameter of tumors,).

＃４ Authors described that “Imaging study-based preoperative diagnostic criteria of HCC are based on the presence of focal hepatic lesions with hyper attenuation in the arterial phase and hypo attenuation in the portal phase on dynamic CT or magnetic resonance imaging (MRI)”. However, the sensitivity and specificity of CT and MRI are different. As in #3, the authors should show the breakdown in each group how to diagnose with enhanced CT, EOB- MRI, both (full-study), or other.

#5

In clinical practice, it is uncommon to treat HCC without finding of HCC in preoperative imaging studies especially in cases not indicated for liver transplantation. In the histopathology in this study, authors had better to show tumor size, differentiation, and pathology diagnosis in detail.

＃6

Cut Off index of WFA+-M2BP is different due to etiology,. Authors should analysis WFA+-M2BP with not all but each etiology.

6. PLOS authors have the option to publish the peer review history of their article (what does this mean?). If published, this will include your full peer review and any attached files.

Reviewer #1: No

Reviewer #2: No

Reviewer #3: No

Reviewer #4: No

Reviewer #5: No

---

## [Author Response · Author response to Decision Letter 0]

22 Aug 2023

Response to reviewers

Thank you for many important suggestions very much. We re-analyzed the data and revised our paper considerably in accordance with your suggestions. We modified the Method, Result and Discussion, and added two tables for improvement of the content.

Major revisions

・Some reviewers pointed that our statistical design was not suitable for negative data of M2BPGi. Considering your suggestions, we added the comparison between group with HCC and without HCC. This is reflected in Table 1 and 3. 

・Some reviewers suggested that we should mentioned more diagnosable marker for HCC than M2BPGi. So, we re-analyzed other tumor markers especially with ROC analyze and revealed AFP was superior to M2BPGi. We added about this to the Result, Discussion, and Table 3.

the Result

Line 148-154: Excluding the group and comparing the existence of histological HCC and total tumor diameter with M2BP index or tumor markers, there was a significant difference only in AFP values with and without HCC. In comparison of total tumor diameter with each data, there was no significant difference in M2BP index and positive correlation between total tumor diameter and AFP, or PIVKA-II. (Table 3) 

the Discussion

Line 210-213: Considering only the presence of tumor or total tumor diameter, rather than this grouping, only AFP had significance in all statistical analysis. This suggests that AFP may be better diagnostic biomarker for occult HCC than WFA+-M2BP.

・There was a suggestion to summarize characteristics of the groups. So, we made new Table 2.

・There was an ambiguity of the definition of occult HCC. We made that clear in the Method, and added the sentences about the relationship between MRI and tumor factors, such as tumor size and pathological differentiation, to the Result and Discussion. 

Line 77-79: All of the tumors detected by CT were detected by MRI. There were the tumors weren’t able to be detected by CT but were detected by MRI. So, we determined tumors weren’t diagnosed by MRI as occult HCC.

Line 107-113: 

Occult HCC diagnosed by EOB-MRI

Two hundred eleven tumors were able to be counted as that were well-circumscribed and measurable in all patients. There were 152 well-differentiated HCCs and 59 moderately-differentiated HCCs. Of 152 well-differentiated hepatocellular carcinomas, 42 tumors were able to be detected by EOB-MRI. Of 59 moderate-differentiated hepatocellular carcinomas, 39 tumors were able to be detected by EOB-MRI. The data for each tumor diameter are shown in the table 1.

Line 168-173: In addition, the size and number of HCC are related to the indication of liver transplantation by the Milan criteria. It is therefore important to diagnose occult HCC for a success outcome of liver transplantation. Diagnosis of occult HCC relies on imaging tests. It is reported that EOB-MRI is superior to MD-CT in detecting small HCC. In liver cirrhosis patients, however, the detectability of HCC is reduced in EOB-MRI due to the reduced contrast effect.

Thanks to your kind suggestions, we believe this revision made the point of our paper much clearer. We hope you will accept our revision.

Review Comments to the Author

●Reviewer #1: 

This paper by Nakagawa et al. clarified the clinical utility of serum Wisteria floribunda agglutinin-positive Mac-2 binding protein (M2BP) in end-stage liver disease. Using the explanted end-stage liver of liver transplant recipients, the authors showed that M2BP is not elevated due to occult hepatocellular carcinoma (HCC). The clinical usefulness of M2BP is evident in diagnosing liver fibrosis grade and assessing the risk of future development of HCC. However, the M2BP utility for evaluating the current hepatocellular carcinoma existence is controversial. This paper clarifies this point and is clinically meaningful. There are several comments from the reviewer.

Major

1. The main result in this paper is a comparison between N (who did not have HCC in the explanted liver) and D or I (who had more HCC than preoperatively diagnosed). However, as the author listed in the limitation section, the number of N groups (n=6) control is small as a control group. This may be a critical problem, so we must treat the results carefully.

→Thank you for pointing it out. We were also concerning about the number of N group. While this data might not have statistical meaning, we compared between N, D, I, DS group after increasing N group to 30 patients. Whole livers of these patients were diagnosed pathologically as non HCC. But these pathological tests weren’t done with the protocol used in this study. Including this data, there was no differences in M2BP between their groups. We show this reference data below. We added the sentence; “It is the same result in the comparison with the addition of the cases that were diagnosed as no HCC with histological test not using this study’s protocol” in the Discussion. There is a reference table in the file of Responce to reviewers.

While it is true that N is scarce, the very fact that a whole liver is removed and pathologically diagnosed in the first place, and that WFA+-M2BP is measured immediately before the removal is a very rare situation in itself, and is the strength of this paper. We know the data has the scarcity and variability, so we used the Wilcoxon rank-sum test for correct statistical test.

2. To show that M2BP is not useful for assessing HCC existence, comparing the difference of M2BP between the pathological stage of HCC diagnosed using the whole slice explanted liver may be helpful.

→Thank you for your suggestion. We re-analyzed as your suggestion. There was no significant difference in WFA+-M2BP index between pathologic stages of hepatocellular carcinoma diagnosed using whole liver resection. We added this data in Table2 newly.

●Reviewer #2: 

This paper was written about WFA-M2BP and HCC. The usefulness to detect HCC that was not shown preoepratively was studied. However, authors did not understand that the value of WFA-M2BP differs by the etiology of underlying liver disease. Therefore, this paper is wrong from the study design. Also, WFA-M2BP is not tumor marker but fibrosis marker and predictive marker.

→Thank you for your important comments. However, the predictive and diagnostic potential of WFA+-M2BP for HCC has been discussed in previous reports. (a-c) We believe that it is natural to evaluate the diagnostic potential based on these previously published papers. In addition, we also included investigation stratified by background hepatic disease, because we also thought that it would depend on the etiology of the underlying liver disease. We added to evaluate fibrosis status considering about the relationship between WFA+-M2BP and liver fibrosis.

a. Yamasaki K, Tateyama M, Abiru S, Komori A, Nagaoka S, Saeki A, et al. Elevated serum levels of Wisteria floribunda agglutinin-positive human Mac-2 binding protein predict the development of hepatocellular carcinoma in hepatitis C patients. Hepatology. 2014;60:1563–1570.

b. Chuaypen N, Chittmittraprap S, Pinjaroen N, Sirichindakul B, Poovorawan Y, Tanaka Y, et al. Serum Wisteria floribunda agglutinin-positive Mac-2 binding protein level as a diagnostic marker of hepatitis B virus-related hepatocellular carcinoma. Hepatol. Res. 2018;48:872–881.

c. Kim SU, Heo JY, Kim BK, Park JY, Kim DY, Han KH, et al. Wisteria floribunda agglutinin-positive human Mac-2 binding protein predicts the risk of HBV-related liver cancer development. Liver Int. 2017;37:879–887.

●Reviewer #3: 

Nagakawa K et al. demonstrated that serum WFA+-M2BP level was not associated with the presence of occult HCC in the explanted liver in liver transplantation. Although it is clinically important to detect HCC before liver transplantation, the authors should consider the following problems.

Major comment

1. The abstract seems confusing because the definition of each group is unclear in the abstract.

→Thank you for pointing it out. Although limited by the number of characters, I think the addition of "histologically" in the definition of grouping made it easier to understand.

2. Many reports has been demonstrated the usefulness of WFA+-M2BP as a predictive marker of HCC development, but not as a diagnostic marker of HCC. The authors should refer to more references which demonstrated usefulness of WFA+-M2BP as a diagnostic marker of HCC.

→Thank you for your comments. That is an important point. Predicted ability was of course clarified, but diagnostic potential was also discussed in previous reports. (a-c) Their discussion drove us to evaluate the diagnostic potential. 

a. Yamasaki K, Tateyama M, Abiru S, Komori A, Nagaoka S, Saeki A, et al. Elevated serum levels of Wisteria floribunda agglutinin-positive human Mac-2 binding protein predict the development of hepatocellular carcinoma in hepatitis C patients. Hepatology. 2014;60:1563–1570.

b. Chuaypen N, Chittmittraprap S, Pinjaroen N, Sirichindakul B, Poovorawan Y, Tanaka Y, et al. Serum Wisteria floribunda agglutinin-positive Mac-2 binding protein level as a diagnostic marker of hepatitis B virus-related hepatocellular carcinoma. Hepatol. Res. 2018;48:872–881.

c. Kim SU, Heo JY, Kim BK, Park JY, Kim DY, Han KH, et al. Wisteria floribunda agglutinin-positive human Mac-2 binding protein predicts the risk of HBV-related liver cancer development. Liver Int. 2017;37:879–887.

3. The diagnostic ability of WFA+-M2BP and the other markers should be analyzed and compared by ROC analysis.

→Thank you for your important suggestion. We re-analyzed the data with the ROC analysis in not only WFA+-M2BP but also other tumor markers. New “table 3” shows these data. At the result, AFP levels had the significance. We added the sentence about that in line 153-154; In addition, the ROC curve analysis showed that the area under the curve of AFP was 0.7757, and that cut-off value was 23.20. (Table 3)

Other additional sentences

Line 99-103: Receiver operating characteristic curves were constructed, and the areas under the curves were compared to test the ability to diagnose occult HCC. Optimal cut-off values for the different scales were determined the Youden’s index, which is the value that maximizes the sum of the sensitivity and specificity.

●Reviewer #4: 

Nakagawa et al. reported that M2BPGi is unsuitable as a diagnostic marker of occult HCC end-stage liver cirrhosis. This paper is interesting in that it provides a detailed analysis of HCC in the liver of a liver transplant recipient. However, it appears that further analysis is needed to determine if M2BPGi can be used to detect occult HCC for analysis.

1) Since it is widely known that M2BPGi is a marker that also reflects fibrosis and inflammation, multiple measurements of M2BPGi seem more necessary for observation of HCC than a single measurement of M2BPGi.

→Thank you for your interesting suggestion. However, since we investigated the relationship between occult HCC diagnosed by whole liver histological examination and the laboratory data at the same point with that, we thought the single measurement of M2BPGi does not miss the point. We will try to investigate that in another study.

2) On the empirical side, AFP, AFP-L3, and DCP may be better at detecting occult HCC. However, we believe that this also makes sense to measure over time.

→Thank you for your important suggestion. We re-analyzed the ability of tumor markers to diagnose the HCC including occult HCC. At the result, AFP levels had the significance. We added it to the Result and mentioned about that in the Discussion. As for the number of measurements, it is as above. We are sorry for not having the data about AFP-L3. We explained that they are one of the new biomarkers for HCC in line 210-213. As for DCP, we investigated it as PIVKA-II in this paper.

The additional sentences;

Line 148-154: Excluding the group and comparing the existence of histological HCC and total tumor diameter with M2BP index or tumor markers, there was a significant difference only in AFP values with and without HCC. In comparison of total tumor diameter with each data, there was no significant difference in M2BP index and positive correlation between total tumor diameter and AFP, or PIVKA-II. (Table 3) 

Line 210-213: Considering only the presence of tumor or total tumor diameter, rather than this grouping, only AFP had significance in all statistical analysis. This suggests that AFP may be better diagnostic biomarker for occult HCC than WFA+-M2BP.

3) HCC is a very heterogeneous group, so if you look at the individual cases, is there anyone who could have predicted occult HCC at all?

→Thank you for pointing it out. It is very difficult to predicting occult HCCs, and we couldn’t predict them in actually. It was precisely your question that motivated our paper. In liver transplantation, the indication for HCC relies on the Milan criteria in that the diagnosis of HCC is depends on imaging tests. We also rely on imaging tests to diagnose HCC and determine appropriate candidacy for liver transplantation. In practice, we couldn’t diagnose some HCCs as this paper. These HCCs are defined as occult HCC. Patients who weren’t able to be predicted HCC were grouped in Detect group that they had occult HCC in. We tried to reveal the ability of predicting and detecting occult HCCs with WFA+-M2BP in this paper. But we weren't able to do that. Though AFP levels in Detect group was higher than No group, there was no significant difference. Looking at the individual cases, some patients had high AFP level. It might be possible to predict occult HCC by using that. But there was no significance between them. In comparison between the presence and absence of HCC, there was the significant difference in AFP level, so we added some sentences about the superiority of AFP. But we didn’t emphasize about it in our paper because of the lack of the significance between Detect and No group. We added this explanation to the Discussion, especially about liver transplantation.

Line 168-173: In addition, the size and number of HCC are related to the indication of liver transplantation by the Milan criteria. It is therefore important to diagnose occult HCC for a success outcome of liver transplantation. Diagnosis of occult HCC relies on imaging tests. It is reported that EOB-MRI is superior to MD-CT in detecting small HCC. In liver cirrhosis patients, however, the detectability of HCC is reduced in EOB-MRI due to the reduced contrast effect.

4) What is the best way to detect occult HCC based on this study by the author or in the literature?

→Thank you for your important suggestion. We re-analyzed our data to investigate the best way to detect occult HCC as you kindly pointed out. It showed that AFP had the significance. We added about that to the Result with new tables, and Discussion. 

They are the sentences added to Result and Discussion.

Line 148-154: Excluding the group and comparing the existence of histological HCC and total tumor diameter with M2BP index or tumor markers, there was a significant difference only in AFP values with and without HCC. In comparison of total tumor diameter with each data, there was no significant difference in M2BP index and positive correlation between total tumor diameter and AFP, or PIVKA-II. (Table 3) 

Line 210-213: Considering only the presence of tumor or total tumor diameter, rather than this grouping, only AFP had significance in all statistical analysis. This suggests that AFP may be better diagnostic biomarker for occult HCC than WFA+-M2BP.

●Reviewer #5: 

K. Nagakawa et al. have reported about utility of serum WFA+-M2BP in occult hepatocellular carcinoma. It is interesting to investigate many patients with liver transplant, however there are serious problems with the study design and logical progress. There are several concerns in this study to be published as a research article.

Major

#1 While the purpose or motivation for the study to determine utility of WFA+-M2BP in occult HCC was understandable, it was ultimately not useful. From the context, it was not a study design to show the negative data of WFA+-M2BP. It would be more logical to reconsider from the viewpoint of finding occult HCC with other significant markers such as AFP, instead of forcing WFA+-M2BP to be the main objective.

→Thank you for your important suggestion. We added new analysis to find more suitable marker for detecting occult HCC in our practice. At the result, AFP may be better for it than WFA+-M2BP. 

They are the sentences added to Result and Discussion.

Line 148-154: Excluding the group and comparing the existence of histological HCC and total tumor diameter with M2BP index or tumor markers, there was a significant difference only in AFP values with and without HCC. In comparison of total tumor diameter with each data, there was no significant difference in M2BP index and positive correlation between total tumor diameter and AFP, or PIVKA-II. (Table 3) 

Line 210-213: Considering only the presence of tumor or total tumor diameter, rather than this grouping, only AFP had significance in all statistical analysis. This suggests that AFP may be better diagnostic biomarker for occult HCC than WFA+-M2BP.

#2 As the authors indicate, it was presumed that WFA+-M2BP was not significant when compared in disease groups at high levels WFA+-M2BP. Authors may investigate to adjust and to stratify by background liver fibrosis status (e.g; chronic hepatitis, / cirrhosis, fibrosis index).

→Thank you for your advice. All patients had cirrhotic liver, and no significant differences were found regarding liver fibrosis factor or hepatic reserve capacity. We added the data to Table 2.

#3 The patients are divided in groups as N, D, I, and DS, however there was no Table which summarize each characteristic. Authors should summarize about characteristics of each group (e.g., age, gender, liver function: Bil, Alb, Fib-4 index, number or diameter of tumors,).

→Thank you for your important suggestion. We made “Table 2” newly. It has the data about not only M2BP and tumor markers but liver fibrosis factor, hepatic reserve capacity, total tumor diameter, and UICC stage.

#4 Authors described that “Imaging study-based preoperative diagnostic criteria of HCC are based on the presence of focal hepatic lesions with hyper attenuation in the arterial phase and hypo attenuation in the portal phase on dynamic CT or magnetic resonance imaging (MRI)”. However, the sensitivity and specificity of CT and MRI are different. As in #3, the authors should show the breakdown in each group how to diagnose with enhanced CT, EOB- MRI, both (full-study), or other.

→Thank you for pointing it out. We tested both of contrast enhanced CT and MRI in all cases and used both to diagnose HCC by imaging. We changed “or” to “and” in line 80 on p. 6. In addition, we added the sentences about the definition of occult HCC in line 77-79, “All of the tumors detected by CT were detected by MRI. There were the tumors weren’t able to be detected by CT but were detected by MRI. So, we determined tumors weren’t diagnosed by MRI as occult HCC.“ We added the result and discussion about the ability of diagnosis of MRI. We suggested the border line about the size of tumor that can be diagnosed MRI statistically in the Result. As for the breakdown you pointed out, we suggested that AFP besides the imaging tests may be better diagnostic marker.

Line 107-113: Occult HCC diagnosed by EOB-MRI

Two hundred eleven tumors were able to be counted as that were well-circumscribed and measurable in all patients. There were 152 well-differentiated HCCs and 59 moderately-differentiated HCCs. Of 152 well-differentiated hepatocellular carcinomas, 42 tumors were able to be detected by EOB-MRI. Of 59 moderate-differentiated hepatocellular carcinomas, 39 tumors were able to be detected by EOB-MRI. The data for each tumor diameter are shown in the table 1.

#5 In clinical practice, it is uncommon to treat HCC without finding of HCC in preoperative imaging studies especially in cases not indicated for liver transplantation. In the histopathology in this study, authors had better to show tumor size, differentiation, and pathology diagnosis in detail.

→Thank you for your suggestion. We re-analyzed about suggested points. We added the sentences about that in line 107-113, and showed the detailed data in Table 1.

#6 Cut Off index of WFA+-M2BP is different due to etiology. Authors should analysis WFA+-M2BP with not all but each etiology.

→Thank you for your comment. Since we have not evaluated M2BP qualitatively, we believe that it is possible and meaningful to examine M2BP numerically regardless of etiology. However, we are also considering the etiology, and are examining the data separately for background liver, such as HBV and HCV.

---

## [Decision Letter · Decision Letter 1]

13 Sep 2023

PONE-D-23-05749R1Serum Wisteria floribunda agglutinin-positive human Mac-2 binding protein is unsuitable as a diagnostic marker of occulthepatocellular carcinoma in end-stage liver cirrhosis.PLOS ONE

Dear Dr. Eguchi,

Thank you for submitting your manuscript to PLOS ONE. After careful consideration, we feel that it has merit but does not fully meet PLOS ONE’s publication criteria as it currently stands. Therefore, we invite you to submit a revised version of the manuscript that addresses the points raised during the review process.

ACADEMIC EDITOR:The overal quality of this manuscript has been improved. However, it is still difficult to understand the impact and utility of the negative clinical data regarding M2BPGi. The authors should emphasize the significance and practical implications of this study for clinical practice in the discussion section.

We look forward to receiving your revised manuscript.

Kind regards,

Jincheng Wang

Academic Editor

PLOS ONE

Journal Requirements:

Additional Editor Comments:

The overal quality of this manuscript has been improved. However, it is still difficult to understand the impact and utility of the negative clinical data regarding M2BPGi. The authors should emphasize the significance and practical implications of this study for clinical practice in the discussion section.

Reviewers' comments:

Reviewer's Responses to Questions

**Comments to the Author**

1. If the authors have adequately addressed your comments raised in a previous round of review and you feel that this manuscript is now acceptable for publication, you may indicate that here to bypass the “Comments to the Author” section, enter your conflict of interest statement in the “Confidential to Editor” section, and submit your "Accept" recommendation.

Reviewer #1: All comments have been addressed

Reviewer #2: All comments have been addressed

Reviewer #3: All comments have been addressed

Reviewer #4: All comments have been addressed

Reviewer #5: (No Response)

2. Is the manuscript technically sound, and do the data support the conclusions?

Reviewer #1: Partly

Reviewer #2: No

Reviewer #3: Yes

Reviewer #4: Yes

Reviewer #5: Partly

3. Has the statistical analysis been performed appropriately and rigorously? 

Reviewer #1: Yes

Reviewer #2: No

Reviewer #3: Yes

Reviewer #4: Yes

Reviewer #5: Yes

4. Have the authors made all data underlying the findings in their manuscript fully available?

Reviewer #1: Yes

Reviewer #2: No

Reviewer #3: Yes

Reviewer #4: Yes

Reviewer #5: Yes

5. Is the manuscript presented in an intelligible fashion and written in standard English?

Reviewer #1: Yes

Reviewer #2: Yes

Reviewer #3: Yes

Reviewer #4: Yes

Reviewer #5: Yes

6. Review Comments to the Author

Reviewer #1: The authors addressed the reviewers' comments properly. Although the sample size is relatively small, the study design that assesses the whole explanted liver is highly original. The reviewer thinks that the paper has a clinical impact on understanding the utility of M2BP.

Reviewer #2: It seems to have been improved with revision. Stratified by the background hepatic disease, the number of each background is too small. Therefore, it is difficult to judge whether it is correct.

Reviewer #3: Nagakawa K et al. demonstrated that serum WFA+-M2BP level was not associated with the presence of occult HCC in the explanted liver in liver transplantation.The author appropriately revised the manuscript according to the reviewer’s comments, and the manuscript is suitable for publication in PLOS one in its present form.

Reviewer #4: Thank you very much for the opportunity to review this manuscript. I have no further comment for this paper.

Reviewer #5: Authors have revised the manuscript. And I confirmed this manuscript has partially improved. However, there is an important point to be clarified.

It is still difficult to understand the impact and utility of the negative clinical data regarding M2BPGi. The authors should emphasize the significance and practical implications of this study for clinical practice in the discussion section. Could you please explain why the authors emphasize M2BPGi in the title (including the short title)?"

7. PLOS authors have the option to publish the peer review history of their article (what does this mean?). If published, this will include your full peer review and any attached files.

Reviewer #1: No

Reviewer #2: No

Reviewer #3: No

Reviewer #4: No

Reviewer #5: No

---

## [Author Response · Author response to Decision Letter 1]

9 Oct 2023

Answers to review comments

Review Comments to the Author

Reviewer #1: The authors addressed the reviewers' comments properly. Although the sample size is relatively small, the study design that assesses the whole explanted liver is highly original. The reviewer thinks that the paper has a clinical impact on understanding the utility of M2BP.

→ Thank you for your kind comments. You pointed out exactly our strength and weakness. However, we believe that our strength of whole liver histological examination is unique and impactful.

Reviewer #2: It seems to have been improved with revision. Stratified by the background hepatic disease, the number of each background is too small. Therefore, it is difficult to judge whether it is correct.

→ Thank you for your pointing it out. We understand the small number of them as referred to in limitation. We would like to keep further study.

Reviewer #3: Nagakawa K et al. demonstrated that serum WFA+-M2BP level was not associated with the presence of occult HCC in the explanted liver in liver transplantation. The author appropriately revised the manuscript according to the reviewer’s comments, and the manuscript is suitable for publication in PLOS one in its present form.

→ Thank you for your kind comments. I’m glad that you accepted our revises.

Reviewer #4: Thank you very much for the opportunity to review this manuscript. I have no further comment for this paper.

→ Thank you very much too. I’m really happy that you accepted our revision.

Reviewer #5: Authors have revised the manuscript. And I confirmed this manuscript has partially improved. However, there is an important point to be clarified. It is still difficult to understand the impact and utility of the negative clinical data regarding M2BPGi. The authors should emphasize the significance and practical implications of this study for clinical practice in the discussion section. Could you please explain why the authors emphasize M2BPGi in the title (including the short title)?"

→ Thank you for your important query. In actually, there were many papers regarding that M2BPGi is a predicting marker of HCC, as mentioned in Discussion. Moreover, Chuaypen et al. literally said M2BPGi represented a sensitive marker of early stage HCC. So, it is reasonable for us to emphasize whether M2BPGi could be a diagnostic marker of occult HCC. We added the below sentences in Discussion to tell why we emphasized M2BPGi in the title.

Line 188-191; In the study, the authors concluded serum WFA+-M2BP represented a sensitive marker of early-stage HCC detection in patients with chronic HBV infection. [4] This conclusion motivated us to investigate WFA+-M2BP as a detecting and diagnostic marker of occult HCC.

[4] Chuaypen N, Chittmittraprap S, Pinjaroen N, Sirichindakul B, Poovorawan Y, Tanaka Y, et al. Serum Wisteria floribunda agglutinin-positive Mac-2 binding protein level as a diagnostic marker of hepatitis B virus-related hepatocellular carcinoma. Hepatol. Res. 2018;48:872–881.

---

## [Decision Letter · Decision Letter 2]

17 Oct 2023

Serum Wisteria floribunda agglutinin-positive human Mac-2 binding protein is unsuitable as a diagnostic marker of occult hepatocellular carcinoma in end-stage liver cirrhosis.

PONE-D-23-05749R2

Dear Dr. Eguchi,

We’re pleased to inform you that your manuscript has been judged scientifically suitable for publication and will be formally accepted for publication once it meets all outstanding technical requirements.

Kind regards,

Jincheng Wang

Academic Editor

PLOS ONE

Additional Editor Comments (optional):

Since all comments have been addressed. This manuscript should be accepted.

Reviewers' comments:

Reviewer's Responses to Questions

**Comments to the Author**

1. If the authors have adequately addressed your comments raised in a previous round of review and you feel that this manuscript is now acceptable for publication, you may indicate that here to bypass the “Comments to the Author” section, enter your conflict of interest statement in the “Confidential to Editor” section, and submit your "Accept" recommendation.

Reviewer #5: All comments have been addressed

2. Is the manuscript technically sound, and do the data support the conclusions?

Reviewer #5: Yes

3. Has the statistical analysis been performed appropriately and rigorously? 

Reviewer #5: Yes

4. Have the authors made all data underlying the findings in their manuscript fully available?

Reviewer #5: Yes

5. Is the manuscript presented in an intelligible fashion and written in standard English?

Reviewer #5: Yes

6. Review Comments to the Author

Reviewer #5: The author responded to the reviewers' comments appropriately. Now this manuscript seems to be acceptable.

7. PLOS authors have the option to publish the peer review history of their article (what does this mean?). If published, this will include your full peer review and any attached files.

Reviewer #5: No

---

## [Editor Report · Acceptance letter]

23 Oct 2023

PONE-D-23-05749R2 

Serum wisteria floribunda agglutinin-positive human Mac-2 binding protein is unsuitable as a diagnostic marker of occult hepatocellular carcinoma in end-stage liver cirrhosis. 

Dear Dr. Eguchi:

I'm pleased to inform you that your manuscript has been deemed suitable for publication in PLOS ONE. Congratulations! Your manuscript is now with our production department. 

Kind regards, 

on behalf of

Dr. Jincheng Wang 

Academic Editor

PLOS ONE